# Current Insights on the Use of Insulin and the Potential Use of Insulin Mimetics in Targeting Insulin Signalling in Alzheimer’s Disease

**DOI:** 10.3390/ijms232415811

**Published:** 2022-12-13

**Authors:** Amy Woodfield, Tatiana Gonzales, Erik Helmerhorst, Simon Laws, Philip Newsholme, Tenielle Porter, Giuseppe Verdile

**Affiliations:** 1Curtin Medical School, Curtin University, Bentley 6102, Australia; 2Curtin Health Innovation Research Institute, Curtin University, Bentley 6102, Australia; 3Centre for Precision Health, Edith Cowan University, Joondalup 6027, Australia; 4Collaborative Genomics and Translation Group, School of Medical and Health Sciences, Edith Cowan University, Joondalup 6027, Australia; 5School of Medical and Health Sciences, Edith Cowan University, Joondalup 6027, Australia

**Keywords:** insulin signalling, Alzheimer’s disease, neurodegeneration, biomarkers, insulin mimetics

## Abstract

Alzheimer’s disease (AD) and type 2 diabetes (T2D) are chronic diseases that share several pathological mechanisms, including insulin resistance and impaired insulin signalling. Their shared features have prompted the evaluation of the drugs used to manage diabetes for the treatment of AD. Insulin delivery itself has been utilized, with promising effects, in improving cognition and reducing AD related neuropathology. The most recent clinical trial involving intranasal insulin reported no slowing of cognitive decline; however, several factors may have impacted the trial outcomes. Long-acting and rapid-acting insulin analogues have also been evaluated within the context of AD with a lack of consistent outcomes. This narrative review provided insight into how targeting insulin signalling in the brain has potential as a therapeutic target for AD and provided a detailed update on the efficacy of insulin, its analogues and the outcomes of human clinical trials. We also discussed the current evidence that warrants the further investigation of the use of the mimetics of insulin for AD. These small molecules may provide a modifiable alternative to insulin, aiding in developing drugs that selectively target insulin signalling in the brain with the aim to attenuate cognitive dysfunction and AD pathologies.

## 1. Introduction

Alzheimer’s disease (AD) is a progressive neurodegenerative disease accounting for approximately 70% of dementia cases worldwide [1]. As described in the 2021 World Alzheimer’s Report, more than 50 million individuals worldwide live with dementia. Projections indicate that the prevalence of dementia is expected to double every 20 years [2]. A greater understanding of the underlying disease mechanisms and the impact of the risk factors of disease progression is of importance in the development of therapeutic or preventative avenues to slow disease progression or reduce risk.

AD pathology is progressive and leads to neurodegeneration and subsequent dementia with an accumulation of its pathological hallmarks occurring decades prior to the onset of cognitive symptoms. One of the earliest events is the accumulation of the soluble neurotoxic oligomers of amyloid-β (Aβ) protein, resulting from its aberrant generation from its precursor molecule, amyloid precursor protein (APP), or impaired clearance. Oligomeric Aβ ultimately aggregates, forming fibrils and accumulates in later stages and forming the neuropathological hallmark amyloid (senile) plaque. In addition to the extracellular Aβ protein pathology, the intracellular tau protein hyperphosphorylates early in the disease process and in later stage aggregates, forming the hallmark neurofibrillary tangles (NFT). Early protein pathologies promote neuroinflammation, which can further drive a vicious cycle of pathological features [3,4]. Other features, including impaired neuronal glucose utilisation, neurovascular dysfunction and brain atrophy, also occur prior to the onset of cognitive symptoms [5]. In vitro and in vivo studies have indicated that small Aβ aggregates (oligomers), rather than amyloid plaques and fibrils, promote synaptic dysfunction [6] with a subsequent accumulation of phosphorylated tau (pTau), correlating with neuronal degeneration and death [7]. A number of known risk factors can influence the accumulation and progression of AD pathologies, ultimately hastening cognitive decline.

Genetic, environmental and lifestyle risk factors have been associated with neurodegeneration in AD. Advanced age and the presence of the *Apolipoprotein E*-*ε4* (*APOEε4*) allele are the major known risk factors for sporadic AD. However, others include education status [8], smoking [8], diet [9], sleep [10], stress [11] and obesity [12]. Type 2 diabetes (T2D) is also recognised as a major AD risk factor [5,13].

T2D is a chronic metabolic disease characterised by peripheral insulin resistance, hyperglycaemia, hyperinsulinemia and widespread inflammation. In later stages, pancreatic β-cell dysfunction becomes apparent with islet amyloid polypeptide deposition, pancreatic inflammation and oxidative stress, ultimately leading to pancreatic β-cell death [14]. Early in disease progression, insulin resistance impacts the peripheral tissues (muscle, liver and adipose tissue), promoting them to become resistant to circulating peripheral insulin. Without the adequate activation of the insulin receptor and the downstream translocation of glucose transporters to the cell surface, glucose is unable to adequately transport into the cell for storage or energy production [15]. Insulin resistance, hyperglycaemia and hyperinsulinemia together promote metabolic dysfunction, peripheral inflammation and oxidative stress [16,17].

Epidemiological, clinical, neuroimaging and biological evidence from human, animal and cellular studies have linked T2D and insulin resistance with the promotion of dementia risk, neurodegeneration and AD pathology (reviewed in [18]). Further, these diseases share a common pathology and pathogenic mechanisms that include insulin resistance, impaired insulin signalling, oxidative stress, inflammation and amyloid deposition (reviewed in [19]). The contribution of insulin resistance and T2D to the progression of cognitive impairment and AD pathology remains to be determined but is beginning to be addressed in well-characterised longitudinal cohort studies [20,21,22,23]. Insulin resistance and T2D have been associated with reduced cognitive functioning, brain atrophy and changes in neuroimaging and cerebrospinal fluid (CSF) AD biomarkers (reviewed in [24]). The neuroimaging of individuals with T2D displays consistent relationships between the diabetic status and brain atrophy [25]. Structural magnetic resonance imaging (MRI) studies further support this trend with diabetes being associated with white matter lesions, cortical and hippocampal brain atrophy, and cerebral infarcts [26,27]. These changes have also been associated with reduced cognition and memory [28,29]. Furthermore, in diabetic individuals, a more rapid progression in brain atrophy is observed compared to that seen in normal brain ageing [30]. In addition, post mortem studies on human brains reported reduced insulin receptor expression in patients with AD [31]. Investigations of the relationship between insulin resistance, T2D, cognition and AD biomarkers, particularly at the early stages prior to cognitive impairment, are ongoing. These studies remain important for determining the timing of interventions or treatments to slow down progression particularly in the presence of insulin resistance or T2D.

The potential of the common treatments for the management of diabetes as therapies for AD has been explored with mixed results in clinical trials. This is true with the common treatments for diabetes, which include metformin and insulin. Understanding when to intervene is clearly a major goal; however, just as important is a greater insight into the mechanisms and the impacts of these treatments on the brain and the cellular pathways they target. In addition to discussing the underlying mechanisms of insulin resistance in the CNS, this review provided an update on the efficacy of antidiabetic drugs (with a focus on insulin) as therapeutics for dementia as well as explored the possibility of the use of novel small-molecule mimetics, including insulin mimetics.

## 2. Mechanisms of Insulin Resistance and Impaired Insulin Signalling in Type 2 Diabetes and the Brain

### 2.1. Insulin Signalling and Insulin Resistance in Type 2 Diabetes

Insulin signalling is an important cellular pathway that allows cells to take up and utilise glucose. The binding of insulin to its receptor is the first important step in enacting downstream signalling events. Insulin itself is an anabolic peptide hormone secreted from the pancreatic β cells (made up of 51 amino acids), which is important for regulating the metabolism throughout the peripheral tissues (as reviewed in [32,33]). Insulin signalling is a highly controlled system managed by intracellular mechanisms in a range of different organs, including the pancreas, liver and vascular tissue (as reviewed in [34]). Insulin binds to the insulin receptor (IR) on the cell membrane, starting a cascade of signalling steps that are important for regulating cellular processes, which include the glucose and lipid metabolism, protein synthesis, cell growth, division and survival [35]. The IR belongs to a large family of tyrosine kinase receptors, and, upon insulin binding, the receptor undergoes autophosphorylation of the intracellular β-subunit [36]. This triggers the phosphorylation of the insulin receptor substrate (IRS), which allows binding to phosphatidylinositol-3-kinase (PI3K) and the production of phosphatidylinositol (3,4,5)-triphosphate (PIP3) [37]. The AKT kinase (also known as protein kinase B) is a central signalling molecule in the insulin pathway, and PIP3 production promotes AKT translocation to the cell membrane [38]. P13K activation promotes the phosphorylation of AKT at two sites: Firstly phosphoinositide-dependent protein kinase 1 (PDK1) phosphorylates AKT at site T308 in the catalytic protein core. Secondly, the mechanistic target of rapamycin (mTOR) complex 2 (mTORC2) phosphorylates AKT at site S473 in the hydrophobic C-terminal [39]. For the full activation of AKT, both T308 and S473 must be phosphorylated. The phosphorylation of the T308 site is thought to directly control AKT activity; however, without the additional phosphorylation at the S473 site, the kinase activity is significantly reduced [40]. The phosphorylation of AKT (pAKT) promotes the translocation of glucose transporters (GLUT) to the plasma membrane and facilitates the transport of glucose into the cell [41]. Phosphorylated AKT acts upon many substrates, including glycogen synthase kinase 3β (GSK3β), the mammalian target of rapamycin complex 1 (mTORC1) and the nuclear factor κ light chain enhancer of B cells (NFκB). This active form of AKT phosphorylates GSK3β at serine 9, inhibiting the GSK3β phosphorylation of the tau protein; in addition to promoting glycogen synthesis, the activation of MTORC1 promotes autophagy pathways and the formation of autophagosomal vesicles, and NFκB inhibition leads to the downregulation of neuroinflammatory cytokine production (Figure 1A). AKT activation is controlled by Pleckstrin homology (PH) leucine-rich repeat protein phosphatases (PHLPP), with PHLPP1 dephosphorylating AKT at T308 and PHLPP2 dephosphorylating AKT at S473. AKT regulates many cellular processes, including the critical glucose and lipid metabolism, where the impairment of insulin signalling pathways has implications in a range of disease pathomechanisms, one of which is T2D.

A primary characteristic of T2D is insulin resistance, where tissues such as skeletal muscle, liver and adipose tissue become resistant to circulating peripheral insulin. Without the adequate activation of the IR and the downstream translocation of glucose transporters to the cell surface, glucose is unable to adequately transport into the cell for storage or energy production [14]. The inability of glucose to transport across the cell membrane promotes hyperglycaemia. In an attempt to compensate for the insulin inaction at the cell surface, the pancreatic β cells further increase insulin secretion, leading to hyperinsulinemia [42]. Insulin resistance, hyperglycaemia and hyperinsulinemia together promote metabolic dysfunction, peripheral inflammation and oxidative stress [16,17,43]. In the later stages of the T2D phenotype, a transition occurs: the pancreatic β cells become dysfunctional and fail to produce adequate volumes of insulin; therefore, the disease phenotype progresses to an insulin-dependent state stemming from pancreatic β-cell death [44].

### 2.2. Insulin Signalling and Dysfunction in the Brain—A Key Contributor to “Type 3 Diabetes”

The central nervous system (CNS) consists of highly glucose-dependent tissues; thus, it is critical that the insulin signalling pathways are functioning normally to promote cognitive function and memory production [45]. The brain is responsible for more than 20% of all glucose consumption in the body when considered relative to total body mass [46]; thus, the transport of glucose to the cells within the brain is essential. This involves a number of different isoforms of GLUT that are present within the different structures and cells within the brain. At the blood–brain barrier (BBB), GLUT1 is the major isoform; however, transport is also supported by GLUT3 and GLUT4 in the small cerebral vessels. In astrocytes, supporting glial cells, the primary transporter is GLUT4; however, it is assisted by GLUT2 and GLUT3 in the astrocytic end feet. In microglia, GLUT1 is the most highly expressed [47], with GLUT5 also being expressed in ramified, resting and amoeboid microglial phenotypes [48]. In neurons, GLUT3 is the primary transporter, with additional transporters being critical for special neuronal functions [48]. Impaired insulin signalling and the subsequent impaired glucose transport in the brain have been implicated in ageing-related disease pathomechanisms, impacting angiogenesis; BBB function; and endothelial, glial and neuronal cell functions [49]. An impaired glucose metabolism is a feature of neurodegenerative diseases such as AD, and, indeed, imaging of the brain glucose metabolism is thought to be one biomarker for assessing the progression of the disease. Together with evidence implicating brain insulin resistance in the disease, the term “Type 3 diabetes (T3D)” has been coined to describe AD. The discussion of AD as “T3D” has been reviewed elsewhere (see recent review [50]). The section below briefly discusses a key contributor to this phenotype, impaired brain insulin signalling, providing context to the use of insulin and the mimetics of insulin to attenuate or prevent the dysfunction of this pathway in AD.

Insulin is thought to be neuroprotective, and it regulates emotions and a number of other cognitive functions, including executive function, attention, learning and memory processing [49]. IR expression differs between the regions of the brain [31], with high IR expression observed in the hippocampus, hypothalamus, cerebellum, amygdala, cerebral cortex and olfactory bulb. The olfactory bulb is the site of the highest concentration of the IR and of insulin, demonstrating the fastest rate of insulin transport to the brain [49]. The origin of the insulin present in the brain is debatable with conflicting suggestions to whether insulin is produced in the brain or whether insulin crosses the BBB via an insulin transporter [51]. During the normal ageing process, BBB breakdown first occurs in at the hippocampus, which is important for memory formation and learning. This breakdown is exacerbated in AD, providing a mechanism to which peripheral homeostatic changes can influence the brain [52]. Though the exact mechanism facilitating the insulin action in the brain is unclear, it is thought that cerebral insulin signalling is similar to that in the periphery [51]. In the brain, there are two primary pathways activated through insulin binding to the neuronal IR: the PI3K-AKT pathway and the RAS-mitogen-activated protein kinase (MAPK) signalling pathway [41]. One difference in cerebral insulin signalling is in the PI3K-AKT signalling pathway where the AKT3 isoform is expressed in the brain compared to AKT1 isoform is expressed in the periphery [53]. The MAPK signalling pathway is important for neurodevelopment, and, hence, dysfunction in this pathway is associated with neurodevelopmental disorders, such as autism spectrum disorder, schizophrenia and attention deficit hyperactivity disorder [54].

Insulin is also critical for the nonmetabolic functions in the brain contributing to cognitive function, and, as such, impaired PI3K-AKT insulin signalling in the brain has been implicated in neurodegenerative diseases, including AD [51]. Figure 1B demonstrates some of the downstream outcomes when insulin signalling is impaired in the brain. In AD, insulin receptor expression reduces as the brain becomes more insulin resistant; in addition, studies suggests that soluble Aβ species may block insulin binding to the receptor [55]. The insulin receptor substrate undergoes phosphorylation at the serine residues under insulin-resistant conditions, modifying the signalling pathway [56]. Downstream, without the activation of AKT, GSK3β remains unphosphorylated, allowing the hyperphosphorylation of intracellular tau and preventing glycogen synthesis [57]. mTORC1 inhibition reduces normal autophagy, and, together with AKT, it promotes NKκB, which, in turn, promotes neuroinflammatory cytokine production and the activation of resident astrocytes and microglia [58]. In addition, Aβ processing is influenced by insulin and insulin signalling through insulin receptor binding [59]. Under insulin-resistant conditions, APP processing via β-APP cleaving enzyme 1 (BACE1) and γ-secretase is upregulated and has been associated with an increased Aβ burden in vitro and in vivo [60]. Soluble Aβ species are substrates for the insulin-degrading enzyme (IDE), which is primarily secreted by the microglia in the cerebral environment. In the AD brain, IDE levels are reduced in studies using IDE knockout mice, indicating an inverse relationship where low IDE expression promotes the Aβ burden in the brain [61].

Extensive work has been undertaken in rodent models to understand the complex interactions between insulin signalling, cognitive function and the AD associated pathological hallmarks. AD transgenic mouse models have reduced glucose tolerances and insulin sensitivities compared to their wild-type littermates, also displaying learning impairments and increased inflammatory cytokines [62]. In a similar AD mouse model, an insulin signalling impairment was recognised months prior to evidence of peripheral insulin resistance; this was reflected in the large changes in the energy metabolism [63]. Tau hyperphosphorylation was increased in an insulin-deficient AD mouse model [64]. One study reported that rapamycin, an mTOR inhibitor, led to a reduced Aβ burden and tau hyperphosphorylation in amyloid precursor protein (APP) presenilin 1 (PS1) (APP/PS1)-expressing AD transgenic mice, which was associated with an increase in the insulin-degrading enzyme (IDE), a known regulator of amyloid pathology in the brain [65]. In a mouse model of T2D and AD using the streptozotocin (STZ) ablation of the pancreatic β cells, mice with both the T2D and AD phenotypes demonstrated worse spatial learning and recognition, impaired glucose tolerances and reduced IDE levels compared to the AD only model [66]. In a high-fat-diet-induced T2D model, reductions in glucose and insulin tolerance were associated with increased AD biomarkers, reduced synaptic plasticity, reduced IR activation, increased IRS-1 phosphorylation, increased inflammation and stress responses in the brain [67,68]. In an AD mouse model, insulin treatment showed the mitigation of both long-term and short-term memory as well as learning impairments when compared to the controls, and this was associated with a reduced Aβ plaque load and hyper-phosphorylated tau [69,70]. Overall, in vivo mouse studies indicate that reductions in insulin production and impairment to insulin signalling can contribute to the AD neurodegenerative phenotype. Human studies have had similar findings, where reduced PI3K signalling was observed in the hippocampus of nondiabetic AD individuals, and this was associated with increased oligomeric Aβ and a decline in episodic and working memory [71]. The combination of both T2D and AD worsened the signalling deficiency where reduced insulin signalling pathways were associated with an enhanced tau pathology [72]. Further, insulin resistance in nondiabetic AD individuals has been shown to be associated with a reduced glucose metabolism and a reduced grey matter volume [73]. The animal and human studies above highlight the contribution of impaired CNS insulin signalling to the progression of AD. Over the last few decades, there have been a plethora of studies that have investigated diabetes therapeutics, including insulin, to attenuate AD pathology and improve cognitive function.

## 3. Repurposing Diabetes Therapeutics for Alzheimer’s Disease

Evidence that T2D and AD share common disease mechanisms has led to the evaluation of antidiabetic drugs for the treatment of AD. A large body of work continues to focus on identifying the neurological changes promoted by antidiabetic therapeutics and on assessing whether this can slow down disease progression in those with mild cognitive impairment (MCI) or AD. The field has explored many antidiabetic medications. Some of the most investigated drugs include metformin, pioglitazone, glucagon-like receptor 1 agonists and insulin.

### 3.1. Metformin, Pioglitazone and Glucagon-like Receptor 1 Agonists—There Are Benefits in In Vivo Animal Studies but Has This Translated to Clinical Benefits?

One of the most prescribed metabolic and diabetic therapeutics is metformin, a biguanide that acts as an insulin sensitiser and an inhibitor of glucose synthesis in the liver. Metformin is a derivative of the naturally occurring guanidine with glucose-lowering properties; hence, the mechanisms of action for metformin are wide and complex. Metformin is thought to act upon the liver, with indications that metformin can lower hepatic glucose production through the inhibition of complex 1 in the mitochondria, leading to the activation of AMP-activated protein kinase (AMPK) signalling and the downstream inhibition of gluconeogenesis (as reviewed in [74]). In contrast, metformin therapy in nondiabetic individuals is associated with an increase in hepatic glucose production; though, this may be a compensatory mechanism due to the increased levels of plasma glucagon that were observed (as reviewed in [75]). Metformin’s mechanisms in the brain are less clear, with work in animals and human trials ongoing. Many in vivo studies in a number of rodent models have indicated that metformin can reduce AD pathology, attenuate neuroinflammation, improve cognition and have neuroprotective properties (recently reviewed in [76]). Some more recent studies have proposed that metformin can activate the chaperone-mediated autophagy pathway associated with the degradation of Aβ peptides [77], reduce neuroinflammatory cell activation and increase AMPK activation in the hippocampus [78], resulting in the benefits observed in the AD models.

Evidence of benefits in human case-control, observational and clinical trial studies have been mixed. Case-control, observational and prospective association studies have reported no significant effect on cognitive impairment [79,80] to the worsening of cognitive performance [81]. Other studies have indicated that short-term use (1–3 years) is associated with the increased risk of developing AD, but a longer term and high dose is associated with no additional risk [82,83]. In contrast, Shi et al. showed that short-term use (<1 year) was associated with a higher risk, with longer-term use (>2 years) being associated with a lower incidence particularly among the elderly [84]. From these studies, it appears that age, duration and dose may influence the association of metformin treatment with improvements in cognition. Two small randomised clinical trials that have evaluated metformin in the cognitively impaired reported mixed results. One trial in MCI individuals reported no clear change in their Alzheimer’s Disease Assessment Scale–Cognitive Subscale (ADAS-Cog) scores or other parameters (glucose utilisation and plasma Aβ) [85]. The other trial in those with nondementia vascular cognitive impairment reported an improvement in their ADAS-Cog scores, executive functioning and verbal learning [86]. It is likely that the underlying disease pathology and the factors mentioned above may need to be considered if metformin is to be evaluated further in clinical trials.

In addition to insulin and metformin, the potential benefits of pioglitazone have been well investigated in AD. Part of the thiazolidinedione family, pioglitazone increases insulin sensitivity by activating the peroxisome proliferator-activated receptor gamma (PPAR-γ), resulting in increased glucose transporter expression. For AD, preclinical animal studies have shown that pioglitazone reduces neuroinflammation ([87,88], Aβ pathology [89,90] and oxidative stress in the brain [89]. AD transgenic mice treated with pioglitazone have an improved spatial memory over the controls [91,92], which is associated with reduced tau phosphorylation [91] or reduced Aβ pathology, glial cell activation and synaptic loss [92].

The benefits observed in these animal studies have been somewhat translated in clinical studies. Observational studies have shown that long-term pioglitazone use is associated with a reduced incidence of dementia in diabetics and nondiabetics [93], whilst a small randomised placebo-controlled clinical trial in cognitively normal individuals showed increases in hippocampal neuronal activation. These benefits, however, have not been translated in trials with MCI and AD participants, particularly those with T2D where, despite reduced insulin levels, no significant improvement was observed in cognitive function [94]. More recently, a phase III clinical trial to assess the efficacy of pioglitazone in protecting against progression to MCI was terminated when a futility analysis revealed that there was no likely benefit in preventing a cognitive decline [95].

More recent interest has been in evaluating glucagon-like receptor (GLP) agonists, as these agents offer the advantage of crossing the BBB and exerting direct neuroprotective effects [96]. GLP agonists such as liraglutide and exendin-4 have been shown to increase insulin sensitivity in the periphery [97,98,99,100] and the brain [101]. The use of GLP 1 receptor agonists, including exenatide, liraglutide, dulaglutide and sitagliptin, have been shown to significantly lower the risk of AD compared to metformin [102]. Liraglutide has shown promise in rodent models, improving frontal cortex insulin receptor expression and associated insulin signalling pathways [103], resulting in reductions in Aβ production and tau pathology [104] and attenuating oxidative stress and neuroinflammation [105]. This GLP-1 agonist has also been shown to reverse memory impairment and synaptic loss in Aβ mouse models [106] with similar findings in nonhuman primate models [101]. Similar findings were observed in AD mouse models in the presence of T2D [107,108], suggesting multiple mechanisms underlying liraglutide’s neuroprotective benefits. Similar findings have been reported on another GLP-1 agonist, exendin-4, as it has been shown to increase brain insulin synthesis [109], to restore cerebral insulin signalling [110], to attenuate Aβ and tau pathology as well as neuroinflammation [111,112,113] and to improve cognition [113]. A dual glucagon-like insulinotropic polypeptide (GIP) has showed promise as a therapeutic agent in AD and Parkinson’s disease (as reviewed in [114]). In an APP/PS1 mouse model and an SHSY5Y neuroblastoma in vitro model, cotreatment with the GIP dual agonist and the GLP receptor agonist liraglutide stimulated neurogenesis in vivo and protected against reactive oxygen species (ROS) in vitro. In addition, GIP alone attenuated neuroinflammation in both APP/PS1 and their wild-type littermates [115].

Despite the repeated benefits in animal models, there have been few human studies evaluating GLP-1 agonists. One 6-month randomised placebo-controlled, double-blind clinical trials of liraglutide have been planned, such as the “Evaluating Liraglutide in Alzheimer’s Disease” (ELAD) study, a 12-month study in participants with mild AD [73] with outcomes based on cognition and brain imaging. Only one small pilot randomised placebo-controlled trial has been reported for exendin-4, where MCI or those with probable AD were administered exenatide for 18 months [116]. Although the drug was well tolerated, there was no significant change in cognition or other biomarker parameters due to the study being underpowered because of early termination by the sponsor.

### 3.2. Is insulin the Most Promising Candidate?

The use of insulin as a therapeutic agent in the brain is not a new concept and has been investigated extensively over the past 3 decades. As described above, insulin resistance and impaired insulin signalling in the CNS have detrimental effects on critical brain function and promote neurodegenerative processes. Furthermore, reduced insulin and insulin-like growth factor (IGF) activity have been observed in AD hippocampal tissue, resulting in downstream IR signalling impairment [72]. Reduced IR activation is correlated with an increased Aβ burden and reduced cognitive function [71]. It is therefore reasonable to suggest that restoring insulin signalling via administering insulin is one strategy to prevent neurodegenerative diseases such as AD. Numerous animal studies have suggested that insulin can exert neuroprotective effects and attenuate AD pathology (Table 1). Similarly, human clinical studies indicate that insulin can improve cognitive function and memory [117,118,119].

Animal studies support the beneficial impacts of insulin therapy; though, as insulin cannot passively transport across the BBB, the majority use an intranasal route (Table 1). This delivery method bypasses the BBB and is rapidly delivered into the CNS, providing clinically relevant doses directly to the brain. The intranasal route also prevents the risk of peripheral hypoglycaemia, which is associated with the chronic administration of insulin [120]. The intranasal delivery has also been utilised in human clinical studies (Table 2). Intranasal insulin delivered at doses of 20 or 40 international units (IU) per day have been shown to provide significant improvements in memory [117,121,122] and mood [121] compared to the placebo control but do not alter peripheral glucose levels [117,121,123]. The intranasal delivery of insulin has been associated with reduced CSF tau levels and increased CSF Aβ42 levels, which is indicative of a greater Aβ clearance from the brain [64,122], and these changes correlate with the reduced neuroinflammatory activation in the hippocampus and cortex [64]. The benefits of intranasal insulin therapy have been shown to continue up to 2 months after treatment ceases, suggesting long-lasting effects on the brain. These shorter-term studies informed long-term clinical trials, including the “Study of Nasal Insulin Fighting Forgetfulness” (SNIFF), which evaluated 20 IU of insulin and the corresponding placebo delivered twice a day for 12 months in individuals with MCI or early AD (*n* = 289). No adverse effects were reported during this trial, indicating treatment safety. However, no cognitive or functional difference between insulin and the placebo were observed [123]. It was suggested that the change in device to deliver insulin intranasally may have accounted for the lack of change in the cognitive outcomes. An analysis of the data from the first 49 participants that used the same device as the previous trials showed a slower decline in their ADAS-Cog scores and their activities of daily living, supporting the argument that a device change may have influenced the trial findings. The clinical benefits were supported by the findings of an improved white matter hyperintensity volume [124] and a CSF anti-inflammatory profile [119] in these individuals. Other influencing factors include the presence of the *APOEε4* allele. Mustapic and colleagues [125] showed that the insulin-induced changes in insulin signalling molecules within neuronal-enriched extracellular vesicles that were isolated from the plasma of the SNIFF trial participants were strongly correlated with the changes in the ADAS-Cog scores, supporting the cerebral activation of insulin signalling. This relationship, however, was only observed in *APOEε4* negative individuals, which was consistent with previous studies which reported that the intranasal insulin effects on cognition are influenced by the *APOEε4* status [126,127]. In addition, the modulatory effects of *APOEε4* may be related to insulin sensitivities, as lower CSF insulin levels and lower insulin sensitivities have been observed in non-*ε4* carriers with AD compared to *ε4* carriers [128,129], and, thus, noncarriers may benefit the most from restoring insulin signalling within the brain. These factors as well as sex, where male (but not female) participants showed benefits with the higher dose of 40 IU of insulin, may influence the efficacy of intranasal insulin and need to be considered as variables in future trials. Indeed, a small phase II trial (*n* = 30, no cognitive impairment) has commenced that compares three different devices for the delivery of intranasal insulin at two doses (20 IU and 40 IU) (URL accessed on 4 March 2022, www.alzforum.org/therapeutics/nasal-insulin).

**Table 1 ijms-23-15811-t001:** Animal studies assessing the effectiveness of insulin treatment in the brain.

Study	Animal Model	Treatment and Insulin Type	Cognitive and Pathological Outcomes
Shingo et al. [130]	STZ induced diabetic Wistar rat	ICV Determir	↑ Spatial learning and memory (MWM). ↓ Hippocampal Aβ burden. ↑ IDE and somatostatin ↑ AKT activation. ↑ Neuronal fibre density.
Maher et al. [131]	Sprague–Dawley rat	IN Human Insulin	No behavioural abnormalities observed. No neurodegenerative pathology observed at 12 weeks.
Apostolatos et al. [69]	C57BL6 mice	IN Human Insulin	↑ Spatial memory (radial arm water maze). ↑ Hippocampal PKCδII expression.
Mao et al. [70]	APP/PS1 mice	IN Human Insulin	↓ Anxiety-related behaviour. Improved spatial learning and memory (MWM). ↑ Brain insulin signalling. ↓ p-JNK activation. ↓ Hippocampal and cortical Aβ deposition. ↓ Soluble Aβ40 and Aβ42 levels. ↑ Hippocampal neurogenesis.
Rajasekar et al. [132]	STZ induced AD Sprague–Dawley rat	IN Human Insulin	↑ Spatial learning and memory (MWM) No effect on locomotion ↑ Cerebral blood flow ↓ ROS and iNOS levels
Gou et al. [133]	STZ induced AD Sprague–Dawley rat	IN Human Insulin	↑ Spatial learning and memory (MWM). ↓ Tau phosphorylation ↓ ERK1/2 and CaMKII phosphorylation ↓ Hippocampal microglial activation ↑ DCX neurogenesis marker ↑ Hippocampal pAKT T308
Chen et al. [134]	STZ induced AD Sprague–Dawley rat	IN Human Insulin	↑ FDG-PET SUVR in the prefrontal and cingulate cortexes. ↑ NeuN and ↓ GFAP in the hippocampus
Fine et al. [135]	6-OHA parkinsonian rats	IN Human Insulin	↑ Motor function ↓ Dopaminergic activity
Jolivalt et al. [136]	Db/db mice STZ model of diabetes	IP Human Insulin	Modest memory improvement. ↓ HbA1c levels Modest changes to IR and GSK3β expression. Modest reduction in tau phosphorylation.
Lochhead et al. [137]	Sprague–Dawley rat	IN Human Insulin	↑ IR phosphorylation.
Rhea et al. [138]	SAMP8 mice	IN Human Insulin	No change in insulin signalling markers.
Fawzy Fahim, et al. [139]	STZ induced diabetic Sprague–Dawley rat	IP Human Insulin	↑ Spatial learning and memory (MWM) ↑ Short-term memory (Y-Maze) ↓ Soluble hippocampal Aβ42
Chen et al. [140]	3 × Tg AD mice	IN Human Insulin	↑ IRβ, PI3K and AKT signalling. ↑ Pre- and postsynaptic markers. ↓ Microglial activation. ↓ Aβ40 in the forebrain.
Stanley et al. [141]	APP/PS1 mice	ICV Human Insulin	↑ AKT phosphorylation.
Kelany et al. [142]	Sprague–Dawley rat	IP Human Insulin	↑ Glycaemic control, oxidative stress and inflammation. ↓ Neuropathy.
Mamik et al. [143]	Feline immunodeficiency virus	IN Human Insulin	↑ Motor speed. ↑ Spatial memory and decision making. ↓ Neuropathy and ↑ neuronal count.

ICV represents intracerebroventricular. IN represents intranasal. IP represents intraperitoneal. IR represents insulin receptor. ↑ represents increased. ↓ represents decreased. Aβ represents amyloid-β. IDE represents insulin-degrading enzyme. HbA1C represents haemoglobin A1c (also known as glycosylated haemoglobin). AKT represents protein kinase B. MWM represents Morris water maze. JNK represents c-Jun N-terminal kinase. APP represents amyloid precursor protein. PS1 represents presenilin 1. STZ represents streptozotocin. PKCδII represents protein kinase C delta II. ROS represents reactive oxygen species. iNOS represents inducible nitric oxide species. ERK represents extracellular-signal-regulated kinase. CaMKII represents calmodulin-dependent protein kinase II. DCX represents doublecortin.

### 3.3. Is There a Therapeutic Potential of Rapid- and Longer-Acting Insulin Analogues for AD?

Although initially purified from animal sources, recombinant sources of human insulin are now the main preparations used for the treatment of diabetes. The generation of biosynthetic human insulin has allowed for modifications, influencing the rate of absorption, duration of action, reproducibility and targeted efficacy in various tissues. A range of insulins are available, including intermediate- and long-acting insulin (for basal) as well as ultrarapid-, rapid- and short-acting preparations (for meal/bolus) or combinations [144]. Rapid-acting insulins, such as glulisine, have demonstrated pharmacodynamics similar to a first-phase insulin release in a normoglycaemic state [145]. Longer-acting insulin analogues, such as detemir and glargine, are considered the “gold standard” for duration, as they mimic the steady 24 h basal endogenous insulin secretion. These second-generation basal insulin analogues have similar a metabolic effect compared to the first-generation analogues; though, second-generation insulins have a lower risk of hypoglycaemia [146].

The intranasal delivery of both rapid- and longer-acting insulin analogues has been evaluated in AD trials (shown in Table 2). The longer-acting insulin analogue detemir has been evaluated with mixed results. A three-week placebo-controlled study in participants with MCI or AD (*n* = 60) with 20 IU or 40 IU of detemir administered intranasally twice daily for three weeks improved the verbal episodic memory of *APOEε4* carriers at the higher dose but worsened the memory of noncarriers; though, no significant change in daily function or executive function was seen [147]. The *APOEε4* dependent effect correlated with insulin resistance in noncarriers, which suggests that, for some individuals, prolonged exposure may promote insulin resistance. This notion is consistent with cohort studies showing elevated insulin levels [148] and diabetes as predictors for the incidence of AD in *APOEε4* negative individuals. A second (pilot) trial evaluated a four-month treatment (40 IU/day) with intranasal detemir or unmodified “regular” human insulin in participants with MCI or AD [122]. Compared to the placebo, the participants that were administered “regular” insulin demonstrated improvements in memory, a preserved brain volume and a reduced CSF tau-P181/Aβ42 ratio. No significant changes in memory or biomarkers were observed following treatment with detemir. Although not well understood, the lack of beneficial effects demonstrated with detemir could be due to several reasons. For example, the chronic administration of long-acting insulins may desensitise the insulin receptor or promote hyperinsulinemia, promoting insulin resistance. Detemir is generated through the acylation of a 14-carbon fatty acid to a lysine in human insulin, resulting in the increased ability to self-aggregate and to bind to albumin. This may mean that detemir may become less effective over time or result in a larger molecule that is difficult to transport across the epithelium of the BBB, even following intranasal delivery, [149] or it may be transported along perivascular channels [122] without entering the brain parenchyma.

Given the demonstrated benefits over detemir, the intranasal delivery of insulin was pushed forward for evaluation as an AD therapeutic agent in subsequent trials. However, there is interest in the assessment of rapid-acting insulin analogues, which promote acute increases in insulin, do not bind to albumin and are rapidly cleared. To date, two fast-acting insulin analogues have been investigated, glulisine (lysine replaced with a glutamic acid at B29, and asparagine replaced with a lysine at position B3) and insulin aspart (proline replaced with an aspartic acid at position B28). A double-blinded randomised crossover study of the glulisine insulin analogue (administered as a single dose on two occasions) failed to show benefits in cognition in *APOEε4* carriers [150]. A longer-duration 6-month phase II trial where participants with MCI or probable AD (*n* = 35) were administered glulisine twice a day reported that, although it was well tolerated, no enhancing effects on cognition, function or mood were observed compared to the placebo [151]. Although the evidence to date suggests that glulisine does not offer any benefits, insulin aspart may show more promise. The intranasal administration of 4 × 40 IU/day of insulin aspart for 8 weeks to healthy men (*n* = 36) demonstrated that their declarative memory improved over that observed with the placebo and insulin [152]. A phase I clinical trial in 24 participants with AD with the administration of 2 × 20 IU/day of insulin aspart for 12 weeks was completed in 2019. The results of this trial are not yet available.

**Table 2 ijms-23-15811-t002:** Human clinical trial outcomes following intranasal insulin intervention in Alzheimer’s Disease.

Study	Phase	Cohort	Dosage Regime	Clinical Outcomes
Benedict et al., 2007 [152]	II	*n* = 36 Males only Cognitively normal	8 weeks 4 × 40 IU per day Insulin aspart	Improved declarative memory compared to placebo.
Claxton et al., 2013 [126]	II	*n* = 104 MCI–moderate AD	4 months 1 × 20 IU or 40 IU per day Insulin humulin	Improved delayed short-term memory in men and women at 20 IU. At 40 IU, only APOEε4 negative men improved.
Rosenbloom et al., 2014 [150]	I	*n* = 9 APOEε4 carriers Mild–moderate AD	Short term 2 × 40 IU Insulin glulisine	Well tolerated. No change in cognition.
Claxton et al., 2015 [147]	II	*n* = 60 MCI–Moderate AD	21 days 1 × 20 IU or 40 IU per day Insulin detemir	Improved memory, including verbal and visuospatial working memory. More effective in those that are highly insulin-resistant and APOEε4 negative.
Craft et al., 2017 [122]	II	*n* = 36 MCI–Moderate AD	4 months 1 × 40 IU per day Insulin humulin or insulin detemir	Insulin humulin: Improved memory. Preserved volumetric MRI. Reduced pTau181/Aβ42 ratio. Insulin detemir: No change compared to placebo.
Craft et al., 2020 [123]	II	*n* = 289 MCI–Moderate AD	12 months 1 × 40 IU per day Insulin humulin	No change in cognitive function. No change in CSF biomarkers. Limited, as 49 participants used a different intranasal device.
Rosenbloom et al., 2021 [151]	II	*n* = 35 MCI–Moderate AD	6 months 2 × 20 IU per day Insulin glulisine	No change in cognitive function. Minor adverse reactions noted.
Kellar et al., 2021 [124]	II	*n* = 78 MCI–Moderate AD	12 months 2 × 20 IU per day Insulin humulin	Improved white matter hyperintensity volume (WMHV). WMHV correlated with worse CSF AD prognostic markers.
Kellar et al., 2022 [119]	II	*n* = 49 MCI–Moderate AD	12 months 2 × 20 IU per day Insulin humulin	Increased anti-inflammatory IFN-γ and eotaxins. Reduced proinflammatory IL-6.

MCI represents mild cognitive impairment. AD represents Alzheimer’s Disease. IU represents international units. CSF represents cerebrospinal fluid. APOE represents apolipoprotein E. MRI represents magnetic resonance image. WMHV represents white matter hyperintensity volume. IFN represents interferon. IL represents interleukin.

## 4. Is There Potential in Evaluating Insulin Mimetics for AD Treatment?

Though critically important for a diabetic individual’s quality of life, insulin therapy is associated with a range of off-target effects, including weight gain [153,154] and hypoglycaemia [155], which can promote cardiovascular dysfunction [156,157], neurological disorders [158,159] and the increased counter secretion of regulatory hormones [160]. These effects are somewhat minimised through the delivery of insulin via the intranasal route rather than peripherally. However, chronic treatment with insulin or its longer-acting higher-potency analogues could contribute to an insulin-resistant state, further exacerbating the underlying disease mechanisms of insulin resistance and promoting disease progression [161,162]. Furthermore, mixed results from trials with detemir and glulisine suggest that creating longer-acting or more potent analogues may not be appropriate for the treatment of cerebral insulin resistance.

One suggestion which may impact the effectiveness of insulin therapeutics is that insulin is considered an amyloidogenic protein (like Aβ) and, hence, has the ability to form oligomers and organise into protofibrils and fibrils (reviewed in [163,164]). The ability of insulin to aggregate and the underlining processes and conditions that promote this (such as temperature or pH) have been recently reviewed elsewhere [165,166,167]. Of clinical relevance is the fact that repeated injections of insulin or the use of an insulin pump can convert insulin into amyloid fibril deposits at the site of administration [168,169]). This can impact the biological activity of insulin and the glycaemic control in diabetes patients [169]. In vitro studies have suggested that insulin aggregates could be cytotoxic. Exposing the neuronal cells to insulin oligomers compromises their integrity, structure and viability, whereas this was not the case for cells exposed to higher-ordered aggregates, such as fibrils [170,171,172]. Whether intermediate aggregates are present and have similar effects in vivo remains to be determined. In addition, the formation of such aggregates following intranasal delivery has not been reported.

Insulin-like molecules have become of interest as potential alternatives to insulin and insulin analogues for the treatment of diabetes, aiming to limit disease exacerbation and off-target side effects whilst promoting normal insulin signalling. Such agents are referred to as insulin mimetics and range from natural extracts to synthetic nonpeptidyl molecules.

### 4.1. Naturally Derived Insulin Mimetics

Naturally derived extracts have shown some mechanistic traits common to insulin and, hence, have been investigated to assess their potential application in T2D. The natural plant alkaloid berberine (BBR) is derived from Chinese goldthread. A meta-analysis assessing randomised controlled trials using the BBR treatment for prediabetes or T2D revealed that BBR improved HbA1c, fasting plasma glucose levels, fasting plasma insulin levels and the measures of insulin resistance (homeostatic modelling assessment of insulin resistance, HOMA-IR) and that, in some cases, it improved inflammatory profiles [173]. BBR was investigated in a STZ rat model of T2D and showed that BBR upregulated GLUT4 expression, which was similar to the results observed with metformin. In db/db mice, the combination of metformin and BBR reduced blood glucose levels and improved insulin sensitivity compared to metformin alone [174]. Further investigation on a primary hepatocyte model treated with the proinflammatory tumour necrosis factor α (TNF-α) has shown that exposure to BBR suppresses the activation of extracellular signal-regulated kinase (ERK) 1/2 and the downstream attenuating serine phosphorylation of IRS-1, therefore allowing the tyrosine phosphorylation of IRS-1 and AKT activation [175]. In type 2 diabetics with dyslipidaemia, the BBR treatment was associated with a decreased fasting plasma glucose, fasting plasma insulin and HOMA-IR in addition to improvements in triglyceride, cholesterol and low-density lipoprotein cholesterol levels compared to the placebo-treated participants [176].

BBR has also been assessed in models of AD. In a 3xTg mouse model of AD, BBR at both 50 mg and 100 mg per kilogram per day improved short- and long-term memory [177,178,179] and reduced Aβ deposition in the cortex and hippocampus by limiting beta-site APP cleaving enzyme 1 (BACE1) expression and APP processing [177,178]. At 100 mg per kilogram per day, BBR improved cerebral blood flow by promoting proangiogenic factors [178], reduced tau phosphorylation at site S396 through the activation of AKT and promoted the downstream inhibitory phosphorylation of GSK3β at S9 [179]. In the TgCRND8 mouse model of AD, the BBR treatment was associated with a reduced Aβ burden and Aβ plaque size in the cortex and hippocampus, reduced microglial and astrocyte neuroinflammatory activation and improved cognitive function at both 25 and 100 mg per kilogram per day [180]. In the APP/PS1 mouse model, 3-month high-dose BBR (260 mg per kilogram per day) improved the endoplasmic reticulum stress in the hippocampus, reduced hippocampal Aβ42 deposition through the downregulation of BACE1 and reduced tau phosphorylation at S202 and S404 through the regulation of GSK3β phosphorylation [181]. In a high-fat-diet rat model of T2D and AD (through a peritoneal STZ injection and a bilateral hippocampal Aβ injection) treated with either 150 mg per kilogram per day of BBR or metformin (positive control) for 28 days, the BBR intervention was associated with a reduced blood glucose, reduced hippocampal neuron loss, increased synaptic integrity, a reduced Aβ burden and improved cognitive function similar to the results of the metformin intervention [182].

Other naturally derived molecules have been investigated to assess their influence on T2D. A single study investigated a raw *Bellis Perennis* (the common daisy) extract developed using ground daisies picked from the local area (Wels, Austria). This extract was compared to traditional insulin and commercial *Bellis Perennis* extracts. The local extract was able to induce GLUT 4 translocation comparable to that of the insulin treatment in vitro. Correspondingly, this extract reduced blood glucose levels in chick embryos [183]. Cinnamaldehyde (Cin), derived from cinnamon, is an important part of Chinese medicines and has shown antidiabetic potential. In STZ induced diabetic rats, Cin reduced the fasting plasma glucose and fasting plasma insulin in addition to promoting GLUT4 expression in muscle and adipose tissues [184]. Pomegranate derivatives have shown broad applications with antioxidant, anti-inflammatory and antidiabetic properties. In the diabetic Zucker rat model, pomegranate seed powder (PSP) showed promising reductions in obesity-induced fatty livers and associated inflammation [185]. In an STZ induced T2D rat model, an intervention with pomegranate juice was associated with reduced plasma cholesterol, triglycerides and low-density lipoproteins compared to the controls [186]. A prospective randomised placebo-controlled trial investigating the influence of PSP on the glucose and lipid metabolism in T2D showed that PSP significantly reduced the fasting blood glucose and haemoglobin A1c (HbA1c, also known as glycosylated haemoglobin) levels compared to the placebo; however, no lipid metabolism measures were significantly impacted [187]. A randomised, double-blind controlled trial investigated the complementary actions of dietary fibre, chia seeds and ginseng extract on the glycaemic control in T2D participants. The outcomes showed favourable trends and were well tolerated; however, only the HbA1c levels showed any statistically significant change following 24 weeks of the intervention [188]. Chiro-inositol (CI), a derivative from the buckwheat plant, has shown insulin mimetic qualities with investigations primarily on its effectiveness in periphery-targeting insulin resistance. A further study on cultured hippocampal neurons showed that the chiro-inositol treatment promoted the activation of insulin signalling pathways. Similar to the native insulin treatment, the CI treatment promoted dendritic IR withdrawal and prevented Aβ induced synaptic damage. No effect was directly observed in Aβ oligomerisation [189].

Although better known for its antioxidant properties, the naturally occurring fatty acid, α-lipoic acid, has been reported to have insulin mimetic activity, binding directly to the tyrosine kinase domain of the IR and activating the PI3K pathway [190]. Present in many foods, including plant-based foods, it is an essential cofactor for a number of enzyme systems and is considered as one of a number of natural products that have potential benefits in AD (as reviewed in [191]). In relation to its insulin mimetic properties in AD, administering α-lipoic acid to an AD mouse model (3 × Tg) has been shown to enhance insulin receptor substrate activation and to subsequently stimulate the PI3K/AKT signalling pathway, promoting glucose uptake [192]. This was associated with improved synaptic plasticity in the mice. In a zymosan (TLR-2 agonist)-induced model of T2D, the 28-day lipoic acid treatment significantly improved insulin sensitivity and cognitive function by improving insulin signalling and reducing inflammatory markers (including interleukin 6, or IL-6) and oxidative stress [193]. The high-fat-fed Wistar rat model displayed cognitive impairment as well as peripheral and hippocampal insulin signalling abnormalities. In this model, the lipoic acid intervention was associated with increased GLUT1 expression in the hippocampus and improved cognitive function and memory [194]. In a high-fat-fed STZ induced rat model of T2D, the 13-week lipoic acid intervention increased the expression of IRS1, PI3K, the pAKT/AKT ratio and the pGSK3β/GSK3β ratio in the hippocampus and promoted increased levels of IDE and improved cognitive function compared to the untreated diabetic mouse model [195]. In humans, a placebo-controlled clinical trial investigated the effectiveness of differing oral lipoic acid doses in T2D participants, and lipoic acid was associated with an enhanced metabolic clearance rate (MCR) of glucose, indicating increased insulin sensitivity. Though positively influencing insulin sensitivity, there were no observed dose-dependent changes in the MCR [196].

### 4.2. Synthetic Small-Molecule Insulin Mimetics

Plant products and natural derivatives are often considered safe alternatives, and several have shown insulin-like action in models of both T2D and AD. Despite this, the non-specificity and further clarity required to determine the active ingredients and potency that are required makes their “drug ability” less attractive. There is, therefore, an equal interest in developing synthetic insulin mimetic molecules. Unlike the insulin analogues, the design of these small molecules typically specifically targets either the insulin receptor or the IGF receptor to reduce potential off-target binding. This is in contrast to native insulin, which can bind to both receptors [197]. In addition, small-molecule mimetic agents have an increased potential to cross the BBB for applications in neurodegenerative diseases, thus reducing the need for intranasal administration which has its own disadvantages, including nasal mucociliary clearance, a limited volume that can be administered all at once, frequent use maybe leading to nasal mucosa damage or irritation and nasal congestion due to colds or allergies affecting drug delivery and potency (reviewed in [198]). In addition, small insulin molecules have a greater potential to be modified to be taken orally.

An example of a small insulin mimetic is compound 1 (Cpd1). This non-peptidyl benzoquinone derivative of the fungal metabolite L-783,281 [199] has been characterised as a highly potent, selective and orally efficacious activator of the IR [200,201]. It directly binds to the α-subunit of the IR, causing autophosphorylation and resulting in the phosphorylation of IRS-1 and activation of PI-3 kinase phosphorylation [199]. The oral administration of Cpd1 to a mouse model of diabetes has been shown to activate the IR tyrosine kinase and to significantly reduce blood glucose levels compared to vehicle-treated mice [199,202]. In male rats, the intracerebroventricular (ICV) administration of Cpd1 was associated with a dose-dependent reduction in food intake and, as a consequence, weight loss was observed [202]. A number of chemical variations have been made to Cpd1 (the alteration or substitution of the third or sixth positions of the parent benzoquinone), resulting in other insulin mimetic derivatives (compounds 2–24) [200]. These compounds have been assessed for the activation of the IR, IGF receptor and ependymal growth factor (EGF) receptor with varying affinities (reviewed in [203]). Of these, Cpd2 and Cpd3 have been assessed in multiple in vivo models of diabetes. Administering Cpd2 to db/db mice and an STZ mouse model of diabetes resulted in the reversal of hyperglycaemia [204]. No such effect was observed with Cpd3 [204]. Similar findings were observed in a high-fat-fed rat model of obesity, where Cpd2 also reduced food intake, resulting in weight loss. Cpd2 treatment was also associated with reduced insulin resistance and adiposity, which was specifically mediated by promoting IR activation [205]. In addition to its glucose-lowering capabilities, Cpd2 promoted glucose uptake and promoted lower plasma fatty acids, triglycerides and leptin levels. Cpd2 also improved cellular insulin signalling in the liver, skeletal muscle and white adipose tissues [206]. Despite these promising in vivo findings, these compounds have not been evaluated in human clinical studies or in vitro or in vivo models of AD or other neurodegenerative diseases. Only one study has investigated the parent molecule L-783,281 as a potential neurotrophin mimic within rat and human primary neuronal cells. Despite activating the neurotrophin Trk receptor, the L-783,281 cytotoxicity profile limited its usefulness as a potential neuroprotective agent [207]. Benzoquinones, such as L-783,281, are generally thought to have limited chronic administration due to their associated cytotoxicity; however, these types of molecules are amenable to modification to be more potent and to be without cytotoxic properties (reviewed in [203]).

In addition to mimetics that activate the IR, those that inhibit the protein tyrosine phosphatases that dephosphorylate the IR and IRS-1 (in effect, promoting insulin signalling) have been investigated in diabetes, with some assessed in AD models. One such molecule is vanadium, a transition-state metal element that has been assessed in both diabetes and AD models. The dyshomeostasis of metal trace elements is a feature in both T2D and AD [208,209,210]. Vanadium predominately exists in its oxidation states, vanadyl (VO^+2^, V^+4^) and vanadate (HVO4, V^+5^). Vanadate has been shown to improve glucose uptake, oxidation and glycogen synthesis in the adipocytes and muscles as well as to increase glycolysis and inhibit gluconeogenesis in the liver with an overall effect of improving glucose homeostasis. Orally administering vanadate to diabetic (db/db) mice and obese (ob/ob) rats has been shown to improve glucose tolerance and homeostasis [211,212]. The benefits in animal models have translated to T2D patients. The oral administration of vanadate and vanadyl improved insulin sensitivity, decreased hyperglycaemia, HbA1c and liver glucose production [213,214]. A more recent trial evaluated the effects of vanadyl daily for 30 months in type 1 diabetics. The small trial showed a 1.6-fold reduction in fasting blood glucose levels [215]. However, in these trials, the levels of the vanadium salts used were associated with bowel irritation and other side effects; in addition, there are concerns that the use of higher levels, due to poor bioavailability, could lead to the accumulation of vanadium in the tissues and could lead to potential toxicity (discussed in [216]). More recently, trials have used an organic vanadium compound synthesised from food additives ethyl maltol and oxovanadium (bis(ethylmalto-lato)oxovanadium(IV), (BEOV)) that displays up to a 3-fold increase in bioavailability following oral administration and a reduced toxicity compared to vanadate salts [217]. BEOV has undergone both phase I, where it was well tolerated, and Phase IIa clinical trials, where it was shown to reduce the fasting blood glucose and HbA1c levels in T2D participants compared to the placebo [218]. However, the subject numbers were very small in the phase II trial (two placebo and seven BEOV), and BEOV (also called AKP-020) was administered for a relatively short period (28 days). Although it showed some promise, the trial was abandoned due to renal problems observed in some patients [217,218].

### 4.3. Perspective on the Future Therapeutic Potential of Insulin Mimetics for AD—From Bench to Bedside

Compared to mimetics, insulin and insulin analogues have progressed further in human clinical trials (detailed in Section 3 above and Table 2). There is an ongoing debate on whether these molecules are still of interest as an alternative to insulin for diabetes treatment, which is mainly driven by their lack of progression through the phases of clinical trials (reviewed in [219]), examples given in Section 4.2. Others include the protein tyrosine phosphatase (PTP) 1B inhibitor, ertiprotafib, which reached phase II trials in the USA and Europe but did not progress due to unsatisfactory, dose-dependent side-effects [220]. Another inhibitor and appetite suppressant, trodusquemine (MSI-1436), was well tolerated in phase I and Ib trials [221]; however, we could not find information on whether the drug progressed. Work to understand how these molecules inhibit PTPs is ongoing, providing a platform to develop similar molecules [222], as they are attractive to pursue based on affinity, selectivity and oral bioavailability.

Despite the work in evaluating the efficacy in T2D, insulin mimetics, such as the vanadium molecule BEOV, have only been recently explored in preclinical in vitro and in vivo AD models. The results look to be promising and appear to attenuate primary AD pathologies (Aβ plaques, hyperphosphorylated tau and neuroinflammation), which offers a multipronged approach to AD therapeutics. In vitro, BEOV has been shown to reduce Aβ levels in neuroblastoma cells, overexpressing APP mutations via decreasing APP levels and the production of Aβ [223]. These findings were translated to in vivo studies with Aβ and tau models. BEOV, administered to amyloid mice (TgAPPswe/PS1Δ9) in their drinking water, ameliorated memory and learning deficits, promoted cerebral glucose uptake, protected hippocampal and cortical synapses and reduced Aβ generation and plaque deposition [223]. Similar findings were observed in the 3xTg model of AD, where BEOV attenuated tau hyperphosphorylation via reducing PTP1B and promoting insulin signalling [224]. BEOV also attenuated neuroinflammatory processes, and it was shown to inhibit proinflammatory NFκ-B signalling, to inhibit microglial and astrocytic activation and to reduce the cerebral levels of proinflammatory cytokines and inducible nitric oxide synthase (iNOS). The anti-inflammatory effects of BEOV were thought to occur via activating PPARγ, where it increased levels, resulting in the repression of NFκ-B signalling [225].

Small-molecule mimetics are more amenable to modifications that can modulate potency, absorption and bioavailability, and, if the intranasal route is used, lower doses can be administered, minimising potential dose related side effects. Importantly, unlike insulin, mimetics may be suitable for oral delivery, which offers an advantage of ease of administration, increasing the likelihood of medication adherence. Their small size also makes them amenable for crossing the BBB, and, thus, improves CNS penetration (a barrier that poses a problem for a number of CNS targeted therapeutics). This will need to be explored further to investigate bioavailability and biodistribution (particularly to and within the brain) in appropriate in vivo models and, if needed, improve these parameters using nano delivery systems. Specificity may also offer another advantage particularly for the synthetically generated nonpeptidyl mimetics, and, indeed, there is ongoing work in creating alternatives to vanadium compounds with improved specificity (reviewed in [226]). All these features make mimetics attractive as an AD therapeutic.

## 5. Concluding Remarks

Insulin signalling is critical for neuronal function, where disruptions to signalling can underpin the pathologies seen in neurodegenerative diseases such as AD. Insulin resistance and impaired insulin signalling are common pathological mechanisms connecting T2D and AD. Thus, commonly used pharmacological approaches in managing insulin resistance and T2D, including metformin, thiazolidinedione and GLP-1 agonists, have been evaluated for use in AD. Despite effectiveness in in vitro and in vivo AD models, consistent evidence of similar benefits in humans is not forthcoming. Administering intranasal insulin has arguably shown the most consistent evidence of efficacy in ameliorating AD pathology and improving cognition in both in vivo and human trials. However, the outcomes from the latest trial have somewhat dampened the enthusiasm for intranasal insulin as a successful treatment for AD. It is clear, however, that multiple factors are influencing the outcomes of the trials, and further investigation has continued in subsequent small pilot studies. A small phase II trial (*n* = 30, no cognitive impairment or MCI) has commenced that compares three different devices for delivering intranasal insulin at two doses (20 IU and 40 IU) to address the discrepancies seen in the SNIFF trial (URL accessed on 4 March 2022, www.alzforum.org/therapeutics/nasal-insulin). The delivery route itself may also pose some issues. The frequent use of the intranasal route may lead to nasal mucosa damage and irritation or may influence mucociliary clearance, leading to congestion. This, in turn, could reduce compliance with using the device and/or limit the effectiveness of delivery. Other factors to consider include temperature, humidification, infections (colds, flus and sinusitis) within the nasal cavity, pre-existing illness or allergies that influence the drug product’s interaction with the nasal mucosa (recently reviewed in greater detail in [227]). Nevertheless, intranasal delivery currently represents the best route for insulin to have direct effects in the brain, and the current interest in developing nano delivery systems for insulin may address some potential issues with current the delivery methods [228].

Considering that an intervention is likely needed prior to the onset of dementia symptoms, before neurodegeneration commences, chronic treatment over decades would be required. The long-term effects of intranasal insulin remain unclear, and whether chronic administration may promote brain insulin resistance and worsen symptoms remains to be determined. Evidence that the effects are maintained after treatment ceases is promising and could be an avenue for managing the duration and number of treatments over the course of an individual’s disease progression. Monitoring with biomarkers would assist this assessment. The use of long-acting insulin analogues, such as detemir, provides another approach to reducing the frequency of treatments; however, trials with detemir to date have not been positive, whereas potential issues have shown a tendency to aggregate and influence the albumin-binding ability, affecting its efficacy. In addition, long-acting insulin analogues, specifically when chronically administered, may lead to the desensitisation of the insulin receptor, thereby leading to a brain-insulin-resistant state. Rapid-acting insulin analogues may provide a more suitable alternative.

The bioavailability, small size, selectivity, ability to be modified and potential to be taken orally make insulin mimetics an exciting alternative to insulin for AD treatment. The investigation of whether mimetics offer benefits for persons living with AD is still in its infancy. However, recent promising results in animal trials with one mimetic (BEOV) warrant the further investigation of insulin mimetics as a viable therapeutic option for AD treatment.

## Figures and Tables

**Figure 1 ijms-23-15811-f001:**
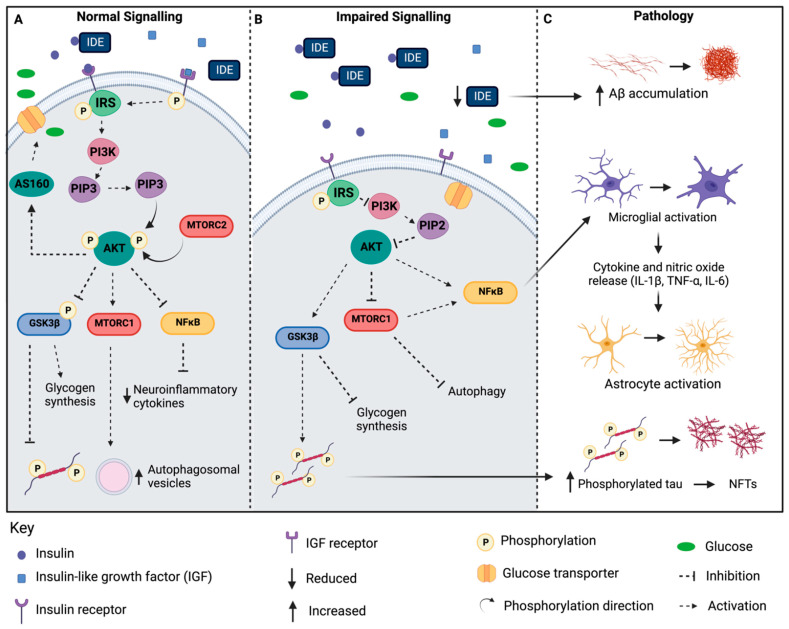
Impaired insulin signalling in Alzheimer’s disease. (**A**) Normal insulin signalling pathway. (**B**) Impaired insulin signalling in the brain. (**C**) Pathological outcomes of impaired insulin signalling in Alzheimer’s disease. Abbreviations: IRS represents insulin receptor substrate. IDE represents insulin-degrading enzyme. PI3K represents phosphoinositide-3-kinase. PIP-2 represents phosphatidylinositol biphosphate. PIP-3 represents phosphatidylinositol 3,4,5 triphosphate. AKT represents protein kinase B. GSK3β represents glycogen synthase kinase 3β. MTORC1 represents mammalian target of rapamycin complex 1. MTORC2 represents mammalian target of rapamycin complex 2. NFκB represents nuclear factor κ light chain enhancer of activated B Cells. IL represents interleukin. TNF represents tumour necrosis factor. NFT represents neurofibrillary tangles. Aβ represents amyloid-β. Created with BioRender.com.

## Data Availability

Not applicable.

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
