# Peer review of "Current Insights on the Use of Insulin and the Potential Use of Insulin Mimetics in Targeting Insulin Signalling in Alzheimer’s Disease"

_ijms, 2022, doi:10.3390/ijms232415811_

Round 1

Reviewer 1 Report

1.       The abstract is not reflecting the text of the manuscript, it may be improved to make it more clear to understand better about the whole manuscript.

2.       APOLIPOPROTEIN E.it may be like normal text, not uppercase

3.       The link between AD and diabetes has already been reported, what is the new insight in this study?

4.       Is there any common target protein that can use to develop the drug? If Yes, it is better to write a separate paragraph.

5.       There are several natural compounds known for these diseases (like Apigenin and several others), so why not the author include this part?

6.       The use of a known drug against another disease and vice versa is termed drug repurposing….so it is better to add about this.

7.       The authors fail to describe the selection of the article (cutoff years). It is not mentioned the selection of all listed references duration.

8.       The manuscript may be improved by adding more recent literature specifically in 2022

Author Response

We thank the reviewer for their comments and feedback on our review, please see the attached word document for a point by point response to their comments. 

Reviewer 2 Report

Revision

Journal   IJMS (ISSN 1422-0067)

Manuscript ID    ijms-1955338

Type   Review

Title   Current insights on the use of insulin and potential use of insulin mimetics in targeting insulin signaling in Alzheimer’s disease.

The Authors discuss the very important aspect of preventing the development of AD by acting on lifestyle risk factors that disrupt the overall functioning of the body through the induction of the metabolic syndrome, and which are currently considered to be the most elevating % of the risk of developing the sporadic form of AD, which accounts for about 99% of all diagnosed AD cases worldwide.

As the Authors emphasized, "Understanding when to intervene is clearly a major goal, however as important is a greater insight into mechanisms and impact of these treatments on the brain and the cellular pathways they target."

Remarkably, the hyperglycemic state diagnosed ambulatory from blood is often too late to introduce effective countermeasures against both the development of T2DM and the activation of the molecular mechanisms of AD. This is due precisely to the preceding impairment of insulin signaling. Insulin is one of the most crucial hormones for normal brain function, and a disruption of its signaling peripherally, is almost immediately reflected by a disruption of pathways dependent on proper binding of insulin to its receptor on the cell membrane surface of brain cells, leading to a host of serious neurological disorders.

The authors synthetically and clearly guide the reader through the complex processes and mechanisms in the introduction, making it easier to further understand the issues discussed.  

Manuscript written in clear, understandable language.

Reviewer Comments:

Major changes:

1.       Line 32 – Reference [1] - Please update with: 2022 Alzheimer's disease facts and figures. Alzheimers Dement. 2022 Apr;18(4):700-789. doi: 10.1002/alz.12638. epub 2022 Mar 14. PMID: 35289055.

2.       Lines 40-41 - I cannot agree with the statement that "One of the earliest events is the accumulation of amyloid-β (Aβ) 40 protein forming the neuropathological hallmark amyloid plaque. "Amyloid plaques are formed quite late in the process of AD pathogenesis, and their formation is preceded in the first place by the formation of products of amyloidogenic proteolysis of APP protein, i.e. Aβ peptides, which already in monomeric form trigger mechanisms leading to the development of AD, and are most toxic in oligomeric form. Senile plaques are formed much later, as a consequence of the disruption of many other processes, among others caused by the appearance of Aβ monomers. Please rephrase

3.       Lines 41-43 – The statement: “Together with neuroinflammation accumulation of Aβ can drive neurofibrillary tangle (NFT) formation (consisting of hyper-phosphorylated tau protein)” is true for the classical amyloid hypothesis of AD development. Currently, some studies suggest that Aβ accumulation does not necessarily precede the formation of NFTs. The dynamics of the development of both these pathologies are likely to be simultaneous in the early stages , before the appearance of the deposits.  Please rephrase.

4.       Line 171 - Please confirm with an appropriate reference that the hippocampal area is unprotected by the BBB. I found no such information in reference #59.

5.       Figure 1 - Figure 1 A shows the dependence of normal inslin signaling on IRS phosphorylation. This is true. However, in Fig. 1B, we no longer see IRS phosphorylation. In fact, in both cases, IRS phosphorylation occurs, only on different residues, where on tyrosine residues the phosphorylation determines proper signaling, and on serine residues it prevents insulin from binding to the receptor for insulin. Please verify the figure and refer to the text accordingly.

6.       Line 247 - Please clarify whether the study was conducted in APP/PS1 or C57BL6/J mice, or comparatively? This is important in terms of the results obtained. A mess in this sentence.

7.       If possible, I suggest refreshing the literature in the references.

Minor changes:

8.       Line 175 – RAS-mitrogen-… Please correct the typo on RAS-mitogen-…

9.       Subchapter 3.1 and later - Please standardize the notation Metformin or metformin?

10.   Subchapter 3.1 and later - Please standardize the notation Exendin 4 or exendin-4?

11.   Subchapter 3.1 and later - Please standardize the notation Liraglutide or liraglutide

12.   Line 457 - Please elaborate on the acronym HbA1c – first use

13.   Line 467 – Please move the expansion of the HOMA-IR abbreviation above to line 458.

14.   Line 471 - Please get the punctuation right this and throughout the text

15.   Line 660 - If you use italics, please standardize throughout the text

16.   Please check for typos throughout the text and in the captions of figures and tables.

17.   Please make sure you always expand abbreviations the first time you use them in the text.

Author Response

(The authors gave the same response as above.)

Reviewer 3 Report

Mechanisms associated with some particular diabetes drugs can help to protect against Alzheimer's disease, Woodfield et al presented an interesting manuscript that provided insight into how targeting insulin signaling in the brain as an alternative therapeutic target for AD, especially insulin mimetics. However, there are some similar approaches that have been recently reviewed (Such as PMID: 2626833; PMID: 35794109; PMID: 33515428...), thus, to make the work more outstanding, several following comments need to be further considered:

1. Since their aim was to review the efficacy of insulin and its analogs with a focus on outcomes in human clinical trials. I would suggest an additional chapter about it, as well as a new table to summarize all the relevant studies at clinical stages so far

2. An additional chapter to discuss diabetes and AD: Mechanisms and insulin resistance/impaired insulin signaling aspects would be nice.

3. Alzheimer's disease is becoming increasingly common, but there are no drugs to affect the course of the disease and the development of new drugs is a slow, costly, and complex process. An alternative strategy is therefore to find already approved drugs that can prove efficacious against the disease and give them a new area of application. Diabetes drugs have been put forward as possible candidates, but so far the studies that have tested diabetes drugs for Alzheimer's disease have not produced convincing results. Thus, further investigation and clinical trials about insulin mimetics for an AD therapeutic are still challenging. I suggest having a perspective chapter to discuss the clinical translation of insulin mimetics from bench to bedside.

Author Response

(The authors gave the same response as above.)

Round 2

Reviewer 2 Report

Dear Authors, 

Thank you for responding to my comments and making appropriate changes to the manuscript. I am satisfied.

I recommend the paper for publication.

I wish you success. 

Author Response

We thank the reviewer for their kind words and good recommendations/comments.

Reviewer 3 Report

Thanks for addressing almost my comments.

1. However, Table 2 (last column) needs to summarize some of the significant findings rather than present what the original described. 

2. Given more recent studies continue to indicate evidence linking T3D with AD, this state-of-the-art aimed to demonstrate the relationship between T3D and AD based on the fact that both the processing of amyloid-β (Aβ) precursor protein toxicity and the clearance of Aβ are attributed to impaired insulin signaling, and that insulin resistance mediates the dysregulation of bioenergetics and progress to AD. The present review would be outstanding if the T3D is included!

3. As I lastly suggested, please make a perspective chapter separately to discuss the clinical translation of insulin mimetics from bench to bedside since a similar topic has been newly published 10.1038/s41380-022-01792-4 

Author Response

Please see the attached word document containing a point-by-point response to the reviewers comments. 
